# Single-Model Uncertainties for Deep Learning

**Natasa Tagasovska**
Department of Information Systems
HEC Lausanne, Switzerland
natasa.tagasovska@unil.ch

**David Lopez-Paz**
Facebook AI Research
Paris, France
dlp@fb.com

## Abstract

We provide single-model estimates of aleatoric and epistemic uncertainty for deep neural networks. To estimate aleatoric uncertainty, we propose Simultaneous Quantile Regression (SQR), a loss function to learn all the conditional quantiles of a given target variable. These quantiles can be used to compute well-calibrated prediction intervals. To estimate epistemic uncertainty, we propose Orthonormal Certificates (OCs), a collection of diverse non-constant functions that map all training samples to zero. These certificates map out-of-distribution examples to non-zero values, signaling epistemic uncertainty. Our uncertainty estimators are computationally attractive, as they do not require ensembling or retraining deep models, and achieve competitive performance.

## 1 Introduction

Deep learning permeates our lives, with prospects to drive our cars and decide on our medical treatments. These ambitions will not materialize if deep learning models remain unable to assess their confidence when performing under diverse situations. Being aware of uncertainty in prediction is crucial in multiple scenarios. First, uncertainty plays a central role on deciding when to abstain from prediction. Abstention is a reasonable strategy to deal with anomalies [11], outliers [39], out-of-distribution examples [69], detect and defend against adversaries [75], or delegate high-risk predictions to humans [14, 27, 12]. Deep classifiers that "do not know what they know" may confidently assign one of the training categories to objects that they have never seen. Second, uncertainty is the backbone of active learning [68], the problem of deciding what examples should humans annotate to maximally improve the performance of a model. Third, uncertainty estimation is important when analyzing noise structure, such as in causal discovery [52] and in the estimation of predictive intervals. Fourth, uncertainty quantification is one step towards model interpretability [3].

Being a wide-reaching concept, most taxonomies consider three sources of uncertainty: approximation, aleatoric, and epistemic uncertainties [20]. First, approximation uncertainty describes the errors made by simplistic models unable to fit complex data (e.g., the error made by a linear model fitting a sinusoidal curve). Since the sequel focuses on deep neural networks, which are known to be universal approximators [15], we assume that the approximation uncertainty is negligible and omit its analysis. Second, aleatoric uncertainty (from the Greek word *alea*, meaning "rolling a dice") accounts for the stochasticity of the data. Aleatoric uncertainty describes the variance of the conditional distribution of our target variable given our features. This type of uncertainty arises due to hidden variables or measurement errors, and cannot be reduced by collecting more data under the same experimental conditions. Third, epistemic uncertainty (from the Greek word *episteme*, meaning "knowledge") describes the errors associated to the lack of experience of our model at certain regions of the feature space. Therefore, epistemic uncertainty is inversely proportional to the density of training examples, and could be reduced by collecting data in those low density regions. Figure 1 depicts our aleatoric uncertainty (gray shade) and epistemic uncertainty (pink shade) estimates for a simple one-dimensional regression example.

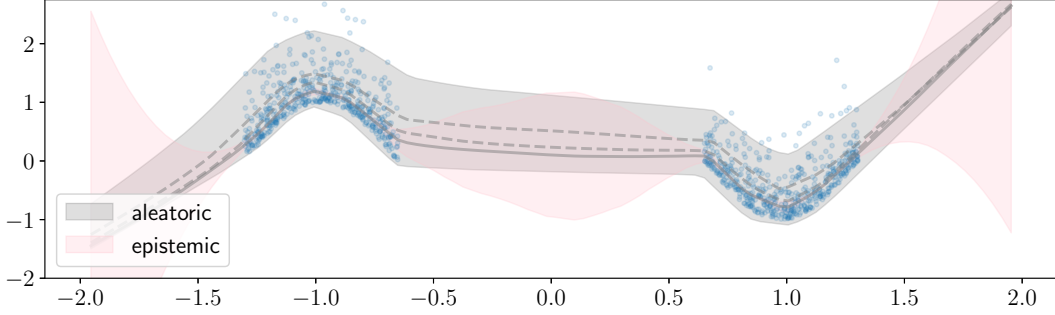

Figure 1: Training data with non-Gaussian noise (blue dots), predicted median (solid line), 65% and 80% quantiles (dashed lines), aleatoric uncertainty or 95% prediction interval (gray shade, estimated by SQR Sec. 2), and epistemic uncertainty (pink shade, estimated by orthonormal certificates Sec. 3).

In general terms, accurate estimation of aleatoric and epistemic uncertainty would allow machine learning models to know better about their limits, acknowledge doubtful predictions, and signal test instances that do not resemble anything seen during their training regime. As argued by Begoli et al. [5], uncertainty quantification is a problem of paramount importance when deploying machine learning models in sensitive domains such as information security [72], engineering [82], transportation [87], and medicine [5], to name a few.

Despite its importance, uncertainty quantification is a largely unsolved problem. Prior literature on uncertainty estimation for deep neural networks is dominated by Bayesian methods [37, 6, 25, 44, 45, 79], implemented in approximate ways to circumvent their computational intractability. Frequentist approaches rely on expensive ensemble models [8], explored only recently [58, 49, 60], are often unable to estimate asymmetric, multimodal, heteroskedastic predictive intervals.

The ambition of this paper is to provide the community with trust-worthy, simple, scalable, single-model estimators of aleatoric and epistemic uncertainty *in deep learning*. To this end, we contribute:

- *Simultaneous Quantile Regression* (SQR) to estimate aleatoric uncertainty (Section 2),
- *Orthonormal Certificates* (OCs) to estimate epistemic uncertainty (Section 3),
- experiments showing the competitive performance of these estimators (Section 4),
- and an unified literature review on uncertainty estimation (Section 5).

We start our exposition by exploring the estimation of aleatoric uncertainty, that is, the estimation of uncertainty related to the conditional distribution of the target variable given the feature variable.

## 2   Simultaneous Quantile Regression for Aleatoric Uncertainty

Let $F(y) = P(Y \leq y)$ be the strictly monotone cumulative distribution function of a target variable $Y$ taking real values $y$. Consequently, let $F^{-1}(\tau) = \inf \{y : F(y) \geq \tau\}$ denote the quantile distribution function of the same variable $Y$, for all quantile levels $0 \leq \tau \leq 1$. The goal of quantile regression is to estimate a given quantile level $\tau$ of the target variable $Y$, when conditioned on the values $x$ taken by a feature variable $X$. That is, we are interested in building a model $\hat{y} = \hat{f}_\tau(x)$ approximating the conditional quantile distribution function $y = F^{-1}(\tau|X = x)$. One strategy to estimate such models is to minimize the pinball loss [23, 48, 47, 22]:

$$\ell_\tau(y, \hat{y}) = \begin{cases} \tau(y - \hat{y}) & \text{if } y - \hat{y} \geq 0, \\ (1 - \tau)(\hat{y} - y) & \text{else.} \end{cases}$$

To see this, write

$$\mathbb{E}\left[\ell_\tau(y, \hat{y})\right] = (\tau - 1) \int_{-\infty}^{\hat{y}} (y - \hat{y}) \mathrm{d}F(y) + \tau \int_{\hat{y}}^{\infty} (y - \hat{y}) \mathrm{d}F(y),$$

where we omit the conditioning on $X = x$ for clarity. Differentiating the previous expression:

$$\frac{\partial \mathbb{E}\left[\ell_\tau(y, \hat{y})\right]}{\partial \hat{y}} = (1 - \tau) \int_{-\infty}^{\hat{y}} \mathrm{d}F(y) - \tau \int_{\hat{y}}^{\infty} \mathrm{d}F(y) = (1 - \tau)F(\hat{y}) - \tau(1 - F(\hat{y})) = F(\hat{y}) - \tau.$$

Setting the prev. to zero reveals the loss minima at the quantile $\hat{y} = F^{-1}(\tau)$. The absolute loss corresponds to $\tau = \frac{1}{2}$, associated to the estimation of the conditional median.

Armed with the pinball loss, we collect a dataset of identically and independently distributed (iid) feature-target pairs $(x_1, y_1), \ldots, (x_n, y_n)$ drawn from some unknown probability distribution $P(X, Y)$. Then, we may estimate the conditional quantile distribution of $Y$ given $X$ at a single quantile level $\tau$ as the empirical risk minimizer $\hat{f}_\tau \in \arg\min_f \frac{1}{n} \sum_{i=1}^n \ell_\tau(f(x_i), y_i)$. Instead, we propose to estimate all the quantile levels simultaneously by solving:

$$\hat{f} \in \arg\min_f \frac{1}{n} \sum_{i=1}^n \mathbb{E}_{\tau \sim U[0,1]} \left[\ell_\tau(f(x_i, \tau), y_i)\right]. \tag{1}$$

We call this model a *Simultaneous Quantile Regression* (SQR). In practice, we minimize this expression using stochastic gradient descent, sampling fresh random quantile levels $\tau \sim \mathcal{U}[0, 1]$ for each training point and mini-batch during training. The resulting function $\hat{f}(x, \tau)$ can be used to compute any quantile of the conditional variable $Y|X = x$. Then, our estimate of aleatoric uncertainty is the $1 - \alpha$ prediction interval ($\alpha$ - significance level) around the median:

$$u_a(x^\star) := \hat{f}(x^\star, 1 - \alpha/2) - \hat{f}(x^\star, \alpha/2). \tag{2}$$

SQR estimates the entire conditional distribution of the target variable. One SQR model estimates all non-linear quantiles jointly, and does not rely on ensembling models or predictions. This reduces training time, evaluation time, and storage requirements. The pinball loss $\ell_\tau$ estimates the $\tau$-th quantile consistently [74], providing SQR with strong theoretical grounding. In contrast to prior work in uncertainty estimation for deep learning [25, 49, 60], SQR can model non-Gaussian, skewed, asymmetric, multimodal, and heteroskedastic aleatoric noise in data. Figure 2 shows the benefit of estimating all the quantiles using a joint model in a noise example where $y = \cos(10 \cdot x_1) + \mathcal{N}(0, \frac{1}{3})$ and $x \sim \mathcal{N}(0, I_{10})$. In particular, estimating the quantiles jointly greatly alleviates the undesired phenomena of *crossing quantiles* [77]. SQR can be employed in any (pre-trained or not) neural network without sacrificing performance, as it can be implemented as an additional output layer. Finally, Appendix B shows one example of using SQR for binary classification. We leave for future research how to use SQR for multivariate-output problems.

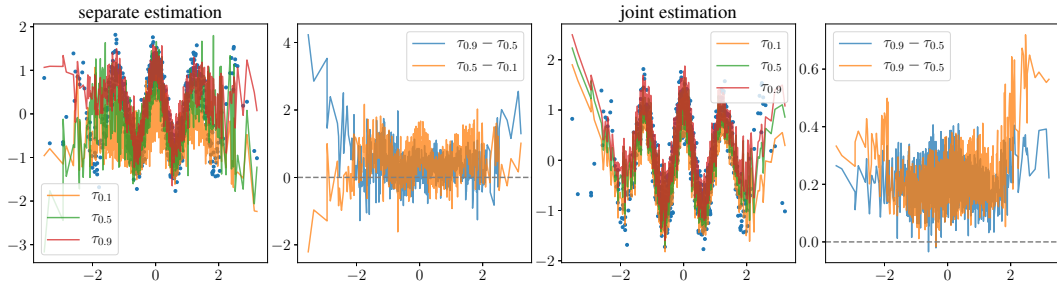

Figure 2: Estimating all the quantiles jointly greatly alleviates the problem of *crossing quantiles*.

## 3 Orthonormal Certificates for Epistemic Uncertainty

To study the problem of epistemic uncertainty estimation, consider a thought experiment in binary classification, discriminating between a positive distribution $P$ and a negative distribution $Q$. Construct the optimal binary classifier $c$, mapping samples from $P$ to zero, and mapping samples from $Q$ to one. The classifier $c$ is determined by the positions of $P$ and $Q$. Thus, if we consider a second binary classification problem between the same positive distribution $P$ and a different negative distribution $Q'$, the new optimal classifier $c'$ may differ significantly from $c$. However, both classifiers $c$ and $c'$ have one treat in common: they map samples from the positive distribution $P$ to zero.

This thought experiment illustrates the difficulty of estimating epistemic uncertainty when learning from a positive distribution $P$ without any reference to a negative, "out-of-domain" distribution $Q$. That is, we are interested not only in one binary classifier mapping samples from $P$ to zero, but in the infinite collection of such classifiers. Considering the infinite collection of classifiers mapping samples from the positive distribution $P$ to zero, the upper tail of their class-probabilities should depart significantly from zero at samples *not* from $P$, signaling high epistemic uncertainty. This intuition motivates our epistemic uncertainty estimate, *orthonormal certificates*.

To describe the construction of the *certificates*, consider a deep model $y = f(\phi(x))$ trained on input-target samples drawn from the joint distribution $P(X, Y)$. Here, $\phi$ is a deep featurizer extracting high-level representations from data, and $f$ is a shallow classifier grouping such representations into classes. Construct the dataset of high-level representations of training examples, denoted by $\Phi = \{\phi(x_i)\}_{i=1}^n$. Second, train a collection of *certificates* $C = (C_1, \ldots, C_k)$. Each training certificate $C_j$ is a simple neural network trained to map the dataset $\Phi$ to zero, by minimizing a loss function $\ell_c$. Finally, we define our estimate of epistemic uncertainty as:

$$u_e(x^\star) := \|C^\top \phi(x^\star)\|^2. \tag{3}$$

Due to the smoothness of $\phi$, the average certificate (3) should evaluate to zero near the training distribution. Conversely, for inputs distinct from those appearing in the training distribution, (3) should depart from zero and signal high epistemic uncertainty. A threshold to declare a new input "out-of-distribution" can be obtained as a high percentile of (3) across in-domain data.

When using the mean squared error loss for $\ell_c$, certificates can be seen as a generalization of distance-based estimators of epistemic uncertainty:

$$u_d(x^\star) = \min_{i=1,\ldots,n} \|\phi(x_i) - \phi(x^\star)\|^2 = \max_{i=1,\ldots,n} 2\phi(x_i)^\top \phi(x^\star) - \|\phi(x_i)\|,$$

which is a set of $n$ linear certificates $C_i^\top \phi(x^\star) = a_i^\top \phi(x^\star) + b_i$ with coefficients fixed by the feature representation of each training example $x_i$. We implement $k$ certificates on top of a $h$-dimensional representation as a single $h \times k$ linear layer, trained to predict the $k$-vector "0" under some loss $\ell_c$. Since we want diverse, non-constant (at zero) certificates, we impose an orthonormality constraint between certificates. Orthonormality enforces a diverse set of Lipschitz-1 certificates. Then, we construct our *Orthonormal Certificates* (OCs) as:

$$\hat{C} \in \arg\min_{C \in \mathbb{R}^{h \times k}} \frac{1}{n} \sum_{i=1}^n \ell_c(C^\top \phi(x_i), 0) + \lambda \cdot \|C^\top C - I_k\|. \tag{4}$$

The use of non-linear certificates is also possible. In this case, the Lipschitz condition for each certificate can be controlled using a gradient penalty [31], and their diversity can be enforced by measuring explicitly the variance of predictions around zero.

Orthonormal certificates are applicable to both regression and classification problems. The choice of the loss function $\ell_c$ depends on the particular learning problem under study. As a rule of thumb we suggest training the certificates with the *same loss function used in the learning task*. When using the mean-squared error, linear orthonormal certificates seek the directions in the data with the least amount of variance. This leads to an interesting interpretation in terms of Principal Component Analysis (PCA). In particular, linear orthonormal certificates minimized by mean squared error estimate the null-space of the training features, the "least-variant components", the principal components associated to the smallest singular values of the training features. This view motivates a second-order analysis that provides tail bounds about the behavior of orthonormal certificates.

**Theorem 1.** *Let the in-domain data follow $x \sim \mathcal{N}(0, \Sigma)$ with $\Sigma = V \Lambda V^\top$, the out-domain data follow $x' \sim \mathcal{N}(\mu', \Sigma')$ with $\Sigma' = V' \Lambda' V'^\top$, and the certificates $C \in \mathbb{R}^{d \times k}$ be the bottom $k$ eigenvectors of $\Sigma$, with associated eigenvalues $\Gamma$. Then,*

$$P\left(\|C^\top x\|^2 - \mathbb{E}[\|C^\top x\|^2] \geq t\right) \leq e^{-t^2/(2\max_j \Gamma_j)},$$
$$P\left(\|C^\top x'\|^2 - \mathbb{E}[\|C^\top x'\|^2] \geq t\right) \leq e^{-t^2/(2\max_j \Lambda_j' \|C_j V_j'\|^2)}.$$

*Proof.* By the Gaussian affine transform, we have that $C^\top x = \mathcal{N}(0, C\Sigma C^\top)$. To better understand the covariance matrix, note $C\Sigma = C\Gamma = \Gamma C \Rightarrow C\Sigma C^\top = \Gamma C C^\top = \Gamma$. The equalities follow by the

eigenvalue definition, the commutativity of diagonal matrices, and the orthogonality of eigenvectors for real symmetric matrices. Then, for in-domain data, we can write $C^\top x = \mathcal{N}(0, \Gamma)$.

Next, recall the Gaussian concentration inequality for $L$-Lipschitz functions $f : \mathbb{R}^k \to \mathbb{R}$, applied to a standard Normal vector $Y = (Y_1, \ldots, Y_k)$, which is $P(f(Y) - \mathbb{E}[f(Y)] \geq t) \leq e^{-t^2/(2L)}$ for all $t > 0$. In our case, our certificate vector is distributed as $\sqrt{\Gamma}Y$, where $Y$ is a standard Normal vector. This means that the function $f(Y) = \|\sqrt{\Gamma}Y\|^2$ has a Lipschitz constant of $\max_j \Gamma_j$ with $j \in 1..k$. Applying the Gaussian concentration inequality to this function leads to the first tail bound.

A similar line of reasoning follows for out-domain data, where $C^\top x' = \mathcal{N}(C\mu', C\Sigma'C^\top)$. The function $f$ applied to a standard Normal vector $Y$ is in this case $f(Y) = \|CC'\sqrt{\Lambda'}Y + C\mu'\|^2$. The Lipschitz constant of this function is $\max_j \Lambda'_j \|C_j C'_j\|^2$. Applying the Gaussian concentration inequality to this function leads to the second result. $\qquad\square$

The previous theorem highlights two facts. On the one hand, by estimating the average in-domain certificate response $\mathbb{E}[\|C^\top x\|^2]$ using a validation sample, the first tail bound characterizes the decay of epistemic uncertainty. On the other hand, the second tail bound characterizes the set of out-domain distributions that we will be able to discriminate against. In particular, these are out-domain distributions where $\|CV'\|$ is small, or associated to a small eigenvalue in $\Lambda'$. That is, certificates will be able to distinguish in-domain and out-domain samples as long as these are drawn from two distributions with a sufficiently different null-space.

**Combined uncertainty**    While the idea of a combined measure of uncertainty is appealing, aleatoric and epistemic unknowns measure different quantities in different units; different applications will highlight the importance of one over the other [19, 44]. We leave for future work the study of combination of different types of uncertainty.

## 4   Experiments

We apply SQR (2) to estimate aleatoric uncertainty in Section 4.1, and OCs (3) to estimate epistemic uncertainty in Section 4.2. For applications of SQR to causal discovery, see Appendix A. Our code is available at `https://github.com/facebookresearch/SingleModelUncertainty`.

### 4.1   Aleatoric Uncertainty: Prediction Intervals

We evaluate SQR (2) to construct $(1 - \alpha)$ Prediction Intervals (PIs) on eight UCI datasets [4]. These are intervals containing the true value about some target variable, given the values for some input variable, with at least $(1 - \alpha)\%$ probability. The quality of prediction intervals is measured by two competing objectives:

- Prediction Interval Coverage Probability (PICP), that is, the number of true observations falling inside the estimated prediction interval;
- Mean Prediction Interval Width (MPIW), that is, the average width of the prediction intervals.

We are interested in calibrated prediction intervals (PICP $= 1 - \alpha$) that are narrow (in terms of MPIW). For sensitive applications, having well calibrated predictive intervals is a priority.

We compare SQR to five popular alternatives. *ConditionalGaussian* [49] fits a conditional Gaussian distribution and uses its variance to compute prediction intervals. *Dropout* [25] uses dropout [38] at testing time to obtain multiple predictions, using their empirical quantiles as prediction intervals. *QualityDriven* [60] is a state-of-the art deep model to estimate prediction intervals, that minimizes a smooth surrogate of the PICP/MPIW metrics. *GradientBoosting* and *QuantileForest* [53] are ensembles of decision trees performing quantile regression.

We use the same neural network architecture for the first three methods, and cross-validate the learning rate and weight decay parameters for the Adam optimizer [46]. For the tree-based of the methods, we cross-validate the number of trees and the minimum number of examples to make a split. We repeat all experiments across 20 random seeds. See Appendix C and code for details.

Table 1 summarizes our experiment. This table shows *test* average and standard deviation PICP of those models achieving a *validation* PICP in $[0.925, 0.975]$. These are the models with reasonably

Table 1: Evaluation of $95\%$ prediction intervals. We show the test average and standard deviation PICP of models achieving a validation PICP in $[0.925, 0.975]$. In parenthesis, we show the test average and standard deviation MPIW associated to those models. We show "none", when the method could not find a model with the desired PICP bounds.

|  | ConditionalGaussian | Dropout | GradientBoostingQR |
|---|---|---|---|
| concrete | $0.94 \pm 0.03\ (0.32 \pm 0.09)$ | none | $0.93 \pm 0.00\ (0.71 \pm 0.00)$ |
| power | $0.94 \pm 0.01\ (0.18 \pm 0.00)$ | $0.94 \pm 0.00\ (0.37 \pm 0.00)$ | none |
| wine | $0.94 \pm 0.02\ (0.49 \pm 0.03)$ | none | none |
| yacht | $0.93 \pm 0.06\ (0.03 \pm 0.01)$ | $0.97 \pm 0.03\ (0.10 \pm 0.01)$ | $0.95 \pm 0.02\ (0.79 \pm 0.01)$ |
| naval | $0.96 \pm 0.01\ (0.15 \pm 0.25)$ | $0.96 \pm 0.01\ (0.23 \pm 0.00)$ | none |
| energy | $0.94 \pm 0.03\ (0.12 \pm 0.18)$ | $0.91 \pm 0.04\ (0.17 \pm 0.01)$ | none |
| boston | $0.94 \pm 0.03\ (0.55 \pm 0.20)$ | none | $0.89 \pm 0.00\ (0.75 \pm 0.00)$ |
| kin8nm | $0.93 \pm 0.01\ (0.20 \pm 0.01)$ | none | none |

|  | QualityDriven | QuantileForest | SQR (ours) |
|---|---|---|---|
| concrete | none | $0.96 \pm 0.01\ (0.37 \pm 0.02)$ | $0.94 \pm 0.03\ (0.31 \pm 0.06)$ |
| power | $0.93 \pm 0.02\ (0.34 \pm 0.19)$ | $0.94 \pm 0.01\ (0.18 \pm 0.00)$ | $0.93 \pm 0.01\ (0.18 \pm 0.01)$ |
| wine | none | none | $0.93 \pm 0.03\ (0.45 \pm 0.04)$ |
| yacht | $0.92 \pm 0.05\ (0.04 \pm 0.01)$ | $0.97 \pm 0.04\ (0.28 \pm 0.11)$ | $0.93 \pm 0.06\ (0.06 \pm 0.04)$ |
| naval | $0.94 \pm 0.02\ (0.21 \pm 0.11)$ | $0.92 \pm 0.01\ (0.22 \pm 0.00)$ | $0.95 \pm 0.02\ (0.12 \pm 0.09)$ |
| energy | $0.91 \pm 0.04\ (0.10 \pm 0.05)$ | $0.95 \pm 0.02\ (0.15 \pm 0.01)$ | $0.94 \pm 0.03\ (0.08 \pm 0.03)$ |
| boston | none | $0.95 \pm 0.03\ (0.37 \pm 0.02)$ | $0.92 \pm 0.06\ (0.36 \pm 0.09)$ |
| kin8nm | $0.96 \pm 0.00\ (0.84 \pm 0.00)$ | none | $0.93 \pm 0.01\ (0.23 \pm 0.02)$ |

well-calibrated prediction intervals; in particular, PICPs closer to 0.95 are best. Among those models, in parenthesis, we show their *test* average and standard deviation MPIW; here, smallest MPIWs are best. We show "none" for methods that could not find a single model with the desired PICP bounds in a given dataset. Overall, our method SQR is able to find the narrowest well-calibrated prediction intervals. The only other method able to find well-calibrated prediction intervals for all dataset is the simple but strong baseline *ConditionalGaussian*. However, our SQR model estimates a non-parametric conditional distribution for the target variable, which can be helpful in subsequent tasks. We obtained similar results for $90\%$ and $99\%$ prediction intervals.

## 4.2 Epistemic Uncertainty: Out-of-Distribution Detection

We evaluate the ability of Orthonormal Certificates, OCs (3), to estimate epistemic uncertainty in the task of out-of-distribution example detection. We consider four classification datasets with ten classes: MNIST, CIFAR-10, Fashion-MNIST, and SVHN. We split each of these datasets at random into five "in-domain" classes and five "out-of-domain" classes. Then, we train PreActResNet18 [34, 51] and VGG [71] models on the training split of the in-domain classes. Finally, we use a measure of epistemic uncertainty on top of the last layer features to distinguish between the testing split of the in-domain classes and the testing split of the out-of-domain classes. These two splits are roughly the same size in all datasets, so we use the ROC AUC to measure the performance of different epistemic uncertainty estimates at the task of distinguishing in- versus out-of- test instances.

We note that our experimental setup is much more challenging than the usual considered in one-class classification (where one is interested in a single in-domain class) or out-of-distribution literature (where the in- and out-of- domain classes often belong to different datasets).

We compare orthonormal certificates (3) to a variety of well known uncertainty estimates. These include Bayesian linear regression (covariance), distance to nearest training points (distance), largest softmax score [35, largest], absolute difference between the two largest softmax scores (functional), softmax entropy (entropy), geometrical margin [80, geometrical], ODIN [50], random network distillation [9, distillation], principal component analysis (PCA), Deep Support Vector Data Description [65, SVDD], BALD [40], a random baseline (random), and an oracle trained to separate the in- and out-of- domain examples. From all these methods, BALD is a Dropout-based ensemble method, that requires training the entire neural network from scratch [26].

Table 2: ROC AUC means and standard deviations of out-of-distribution detection experiments. All methods except BALD are single-model and work on top of the last layer of a previously trained network. Each BALD requires ensembling predictions and training the neural network from scratch.

|  | cifar | fashion | mnist | svhn |
|---|---|---|---|---|
| covariance | $0.64 \pm 0.00$ | $0.71 \pm 0.13$ | $0.81 \pm 0.00$ | $0.56 \pm 0.00$ |
| distance | $0.60 \pm 0.11$ | $0.73 \pm 0.10$ | $0.74 \pm 0.10$ | $0.64 \pm 0.13$ |
| distillation | $0.53 \pm 0.01$ | $0.62 \pm 0.03$ | $0.71 \pm 0.05$ | $0.56 \pm 0.03$ |
| entropy | $0.80 \pm 0.01$ | $0.86 \pm 0.01$ | $0.91 \pm 0.01$ | $0.93 \pm 0.01$ |
| functional | $0.79 \pm 0.00$ | $0.87 \pm 0.02$ | $0.92 \pm 0.01$ | $0.92 \pm 0.00$ |
| geometrical | $0.70 \pm 0.11$ | $0.66 \pm 0.07$ | $0.75 \pm 0.10$ | $0.77 \pm 0.13$ |
| largest | $0.78 \pm 0.02$ | $0.85 \pm 0.02$ | $0.89 \pm 0.01$ | $0.93 \pm 0.01$ |
| ODIN | $0.74 \pm 0.09$ | $0.84 \pm 0.00$ | $0.89 \pm 0.00$ | $0.88 \pm 0.08$ |
| PCA | $0.60 \pm 0.09$ | $0.57 \pm 0.07$ | $0.64 \pm 0.06$ | $0.55 \pm 0.03$ |
| random | $0.50 \pm 0.00$ | $0.51 \pm 0.00$ | $0.51 \pm 0.00$ | $0.50 \pm 0.00$ |
| SVDD | $0.52 \pm 0.01$ | $0.54 \pm 0.03$ | $0.59 \pm 0.03$ | $0.51 \pm 0.01$ |
| BALD$^{\dagger}$ | $0.80 \pm 0.04$ | $\mathbf{0.95 \pm 0.02}$ | $\mathbf{0.95 \pm 0.02}$ | $0.90 \pm 0.01$ |
| **OCs (ours)** | $\mathbf{0.83 \pm 0.01}$ | $0.92 \pm 0.00$ | $\mathbf{0.95 \pm 0.00}$ | $\mathbf{0.91 \pm 0.00}$ |
| unregularized OCs | $0.78 \pm 0.00$ | $0.87 \pm 0.00$ | $0.91 \pm 0.00$ | $0.88 \pm 0.00$ |
| oracle | $0.94 \pm 0.00$ | $1.00 \pm 0.00$ | $1.00 \pm 0.00$ | $0.99 \pm 0.00$ |

Table 2 shows the ROC AUC mean and standard deviations (across hyper-parameter configurations) for all methods and datasets. In these experiments, the variance across hyper-parameter configurations is specially important, since we lack of out-of-distribution examples during training and validation. Simple methods such as functional, largest, and entropy show non-trivial performances at the task of detecting out-of-distribution examples. This is inline with the results of [35]. OCs achieve the overall best average accuracy (90%, followed by the entropy method at 87%), with little or null variance across hyper-parameters (the results are stable for any regularization strength $\lambda > 1$). OCs are also the best method to reject samples across datasets, able to use an MNIST network to reject 96% of Fashion-MNIST examples (followed by ODIN at 85%).

## 5 Related Work

**Aleatoric uncertainty**    Capturing aleatoric uncertainty is learning about the conditional distribution of a target variable given values of a input variable. One classical strategy to achieve this goal is to assume that such conditional distribution is Gaussian at all input locations. Then, one can dedicate one output of the neural network to estimate the conditional variance of the target via maximum likelihood estimation [57, 44, 49]. While simple, this strategy is restricted to model Gaussian aleatoric uncertainties, which are symmetric and unimodal. These methods can be understood as the neural network analogues of the aleatoric uncertainty estimates provided by Gaussian processes [63].

A second strategy, implemented by [60], is to use quality metrics for predictive intervals (such as PICP/MPIW) as a learning objective. This strategy leads to well-calibrated prediction intervals. Other Bayesian methods [33] predict other uncertainty scalar statistics (such as conditional entropy) to model aleatoric uncertainty. However, these estimates summarize conditional distributions into scalar values, and are thus unable to distinguish between unimodal and multimodal uncertainty profiles.

In order to capture complex (multimodal, asymmetric) aleatoric uncertainties, a third strategy is to use implicit generative models [55]. These are predictors that accept a noise vector as an additional input, to provide multiple predictions at any given location. These are trained to minimize the divergence between the conditional distribution of their multiple predictions and the one of the available data, based on samples. The multiple predictions can later be used as an empirical distribution of the aleatoric uncertainty. Some of these models are conditional generative adversarial networks [54] and DiscoNets [7]. However, these models are difficult to train, and suffer from problems such as "mode collapse" [29], which would lead to wrong prediction intervals.

A fourth popular type of non-linear quantile regression techniques are based on decision trees. However, these estimate a separate model per quantile level [77, 86], or require an a-priori finite

discretization of the quantile levels [81, 64, 10, 85]. The one method able to estimate all non-linear quantiles jointly in this category is Quantile Random Forests [53], outperformed by SQR.

Also related to SQR, there are few examples on using the pinball loss to train neural networks for quantile regression. These considered the estimation of individual quantile levels [83, 78], or unconditional quantile estimates with no applications to uncertainty estimation [16, 59].

**Epistemic uncertainty** Capturing epistemic uncertainty is learning about what regions of the input space are unexplored by the training data. As we review in this section, most estimates of epistemic uncertainty are based on measuring the discrepancy between different predictors trained on the same data. These include the seminal works on bootstrapping [21], and bagging [8] from statistics. Recent neural network methods to estimate epistemic uncertainty follow this principle [49].

Although the previous references follow a frequentist approach, the strategy of ensembling models is a natural fit for Bayesian methods, since these could measure the discrepancy between the (possibly infinitely) many amount of hypotheses contained in a posterior distribution. Since exact Bayesian inference is intractable for deep neural networks, recent years have witnessed a big effort in developing approximate alternatives. First, some works [6, 37] place an independent Gaussian prior for each weight in a neural network, and then learn the means and variances of these Gaussians using backpropgation. After training, the weight variances can be used to sample diverse networks, used to obtain diverse predictions and the corresponding estimate of epistemic uncertainty. A second line of work [25, 24, 44] employs dropout [38] during the training and evaluation of a neural network as an alternative way to obtain an ensemble of predictions. Since dropout has been replaced to a large extent by batch normalization [42, 34], Teye et al. [79] showed how to use batch normalization to obtain ensembles of predictions from a single neural network.

A second strategy to epistemic uncertainty is to use data as a starting point to construct "negative examples". These negative examples resemble realistic input configurations that would lay outside the data distribution. Then, a predictor to distinguish between original training points and negative examples may be used to measure epistemic uncertainty. Examples of these strategy include noise-contrastive estimation [32], noise-contrastive priors [33], and GANs [30].

In machine learning literature, the estimation of epistemic uncertainty is often motivated in terms of detecting out-of-distribution examples [69]. However, the often ignored literature on anomaly/outlier detection and one-class classification can also be seen as an effort to estimate epistemic uncertainty [61]. Even though one-class classification methods are implemented mostly in terms of kernel methods [66, 67], there are recent extensions to leverage deep neural networks [65].

Most related to our orthonormal certificates, the method deep Support Vector Data Description [65, SVDD] also trains a function to map all in-domain examples to a constant value. However, their evaluation is restricted to the task of one-class classification, and our experiments showcase that their performance is drastically reduced when the in-domain data is more diverse (contains more classes). Also, our orthonormal certificates do not require learning a deep model end-to-end, but can be applied to the last layer representation of any pre-trained network. Finally, we found that our proposed diversity regularizer (4) was crucial to obtain a diverse, well-performing set of certificates.

## 6   Conclusion

Motivated by the importance of quantifying confidence in the predictions of deep models, we proposed simple, yet effective tools to measure aleatoric and epistemic uncertainty in deep learning models. To capture aleatoric uncertainty, we proposed Simultaneous Quantile Regression (SQR), implemented in terms of minimizing a randomized version of the pinball loss function. SQR estimators model the entire conditional distribution of a target variable, and provide well-calibrated prediction intervals. To model epistemic uncertainty, we propose the use of Orthonormal Certificates (OCs), a collection of diverse non-trivial functions that map the training samples to zero.

Further work could be done in a number of directions. On the one hand, OCs could be used in active learning scenarios or for detecting adversarial samples. On the other hand, we would like to apply SQR to probabilistic forecasting [28], and extreme value theory [18].

We hope that our uncertainty estimates are easy to implement, and provide a solid baseline to increase the robustness and trust of deep models in sensitive domains.

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
