[Supplementary Material]

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

# A  SQR in causal inference applications

## A.1  Causal discovery

We consider the problem of bivariate causal discovery: given a sample from the joint distribution $P(X, Y)$, determine whether $X$ causes $Y$ ($X \rightarrow Y$), or $Y$ causes $X$ ($X \leftarrow Y$). The problem of causal discovery is one where the shape of the conditional distribution of the effect given the cause plays a major role [56]. Indeed, many causal discovery methods can be understood as preferring the "simplest" factorization of the joint distribution (amongst $P(X, Y) = P(Y|X)P(X)$ indicating $X \rightarrow Y$ and $P(X, Y) = P(X|Y)P(Y)$ indicating $X \leftarrow Y$) as the causal explanation [56]. Here, simplest is measured in terms of some measure such as the Kolmogorov complexity [43].

Recently, Tagasovska et al. [76] showed that the pinball loss can be used as a proxy to estimate the Kolmogorov complexity of a probability distribution. Thus, we may interpret the regression model with lower overall pinball loss (amongst the one mapping $X$ to $Y$, and the one mapping $Y$ to $X$) as the causal model.

The high accuracy rates in Table 3 show that our Simultaneous Quantile Regression (SQR, Equation 1) achieves competitive performance at the task of detecting the causal relationship between two variables across all datasets. $m$ represents the number of quantile levels pooled for getting the final causal score. Our evaluation is performed across multiple datasets, including the well-known real-world benchmark from Tübingen [56]. The other datasets are generated according to [76, Section 4], including datasets generated from Additive Noise (AN-), Multiplicative Noise (MN-) and Location-Scale (LS-) models. We compare SQR against a variety of common algorithms in the bivariate cause-effect literature, including LINGAM [70], GR-AN [36], ANM [41], PNL [84], IGCI [17], GPI [73], and EMD [13]. The same table shows the importance of aggregating the pinball loss across multiple quantile levels ($m = 1, 3, 5$) when determining the causal direction between two variables using our proposed SQR. Although using SQR for causal discovery is inspired by Tagasovska et al. [76], their method QCCD is based on non-parametric copulas which are difficult to scale to large number of samples or dimensions. Moreover, SQR is learned end-to-end and is more scalable.

## A.2  Heterogeneous treatment effects

Here we apply simultaneous quantile regression to characterize the conditional quantile distribution $F^{-1}(E|T, X)$ of an effect variable $E$ given some treatment variable $T$ and individual covariates $X$, also known as Quantile Treatment Effect (QTE) estimation [1].

Figure 3 showcases such application for the Student-Teacher Achievement Ratio dataset [2], where we estimate the effect of "class size" (treatment) over the "exam scores" (effect) of students, while considering additional individual features from both the students and the teachers (individual covariates). As independently confirmed by other studies [62], we can also notice the heterogeneity in the

Table 3: Results of causal discovery experiments for all methods (rows) and datasets (columns).

|  | AN | AN-S | LS | LS-S | MN-U | Tübingen |
|---|---|---|---|---|---|---|
| LINGAM | 0.00 | 0.04 | 0.07 | 0.03 | 0.00 | 0.42 |
| GR-AN | 0.05 | 0.15 | 0.11 | 0.20 | 0.50 | 0.50 |
| biCAM | 1.00 | 1.00 | 1.00 | 0.53 | 0.86 | 0.58 |
| ANM | 1.00 | 1.00 | 0.62 | 0.09 | 0.03 | 0.59 |
| GPI | 0.95 | 0.11 | 0.91 | 0.54 | 0.90 | 0.63 |
| IGCI-g | 0.98 | 0.98 | 0.99 | 0.94 | 0.74 | 0.64 |
| EMD | 0.36 | 0.33 | 0.60 | 0.42 | 0.83 | 0.68 |
| IGCI-u | 0.41 | 0.27 | 0.63 | 0.37 | 0.07 | 0.72 |
| PNL | 0.96 | 0.63 | 0.91 | 0.44 | 0.66 | 0.73 |
| QCCD (m=3) | 1.00 | 0.81 | 1.00 | 0.98 | 0.99 | 0.77 |
| SQR (m=1) | 0.99 | 0.51 | 0.98 | 0.78 | 0.62 | 0.75 |
| SQR (m=3) | 0.99 | 0.73 | 1.00 | 0.85 | 1.00 | 0.75 |
| SQR (m=5) | 0.99 | 0.76 | 1.00 | 0.82 | 1.00 | 0.77 |

Figure 3: Results on Heterogeneous quantile treatment effect of "class size" over "exam scores".

treatment effect, namely that the variable "class size" has a larger causal effect for students associated to a larger "exam scores" variable. This shows that uncertainty quantification has a role in policy design.

## B  Quantile classification

Conditional Quantile regression can be recasted to classification method when the target variable is ordinal. In the case of $Y \in \{1, 2...k\}$, one still has the CDF is fully specified by $p(F_Y(y)) = P(Y \leq y|X = x)$, where $y \in k : P(Y = k) > 0$. We next exemplify the application of pinball loss in the case of binary classification for the age of Abalone (UCI dataset).

The Abalone dataset consists of 4177 observations, with 8 features, and we clean the data from missing records. The goal is predicting the age of abalone (number of `Rings`) from some physical measurements. The `Rings` variable was split into two classes: (0 Young) Age $= 0 \,|\, \text{Rings} \leq 10$ and (1 Old) Age $= 1 \,|\, \text{Rings} > 10$. We use the median (in this case also corresponding to $Rings = 10$). To evaluate the results we compare in terms of classification accuracy and ROC AUC against same network architecture and parameters (2 layers with 100 units each, lr $=$ wd $= 1e-3$) trained with Binary Cross Entropy loss. We repeated the experiment 30 times and the results are as follows:

- Accuracy: Pinball $0.78 \pm 0.02$, BCE $0.79 \pm 0.02$
- ROC AUC: Pinball $0.84 \pm 0.03$, BCE $0.87 \pm 0.02$

The results from Pinball loss being within the margin of those from logistic loss, confirm that our method can also be used for classification purposes. Furthermore, SQR provides confidence intervals about the obtained prediction score.

## C  Hyperparameters for experiments

### C.1  Prediction Intervals experiment

For all network-based baselines the results were cross-validated over the following grid:

- learning rate $\in \{1e-2, 1e-3, 1e-4\}$
- weight decay $\in \{0, 1e-3, 1e-2, 1e-1, 1\}$
- dropout rate $\in \{0.1, 0.25, 0.5, 0.75\}$,
- number of epochs 5000
- 20 random seeds

For the rest of the methods for the same 20 random seed, the following parameters per baseline were used:

- GradientBoostingQR: number of learners $\in \{100, 1000\}$ , max depth $\in \{3, 9, 10\}$, learning rate $\in \{1e^{-2}, 1e^{-3}, 1e^{-4}\}$, min samples per leaf $\{3, 9, 15\}$, min samples per split $\{3, 9, 15\}$
- QuantileForest: number of learners $\in \{100, 1000\}$ , min samples per split $\{3, 9, 15\}$, max depth $\in \{3, 5\}$

## C.2 Out-of-distribution experiment

The following parameters per baseline were used:

- covariance: $\lambda \in \{1e^{-5}, 1e^{-3}, 1e^{-1}, 1\}$
- distance: $\text{dist}_{\text{percentile}} \in \{0, 1, 10, 50\}$
- certificates: $c_k \in \{100, 1000\}$ , $c_{\text{epochs}} \in \{10, 100\}$, $c_\lambda \in \{0, 1, 10\}$
- distillation $c_k \in \{100, 1000\}$ , $c_{\text{epochs}} \in \{10, 100\}$
- entropy temperature $\in \{1, 2, 10\}$
- functional $logits \in \{0, 1\}$, temperature $\in \{1, 2, 10\}$
- geometrical $logits \in \{0, 1\}$, temperature $\in \{1, 2, 10\}$
- largest $logits \in \{0, 1\}$, temperature $\in \{1, 2, 10\}$
- odin $\epsilon \in \{0.014, 0.0014, 0.00014\}$, temperature $\in \{100, 1000, 10000\}$
- svdd $c_k \in \{100, 1000\}$ , $c_{\text{epochs}} \in \{10, 100\}$
- pca $c_k \in \{1, 10, 100\}$
- bald $n_{drop} \in \{100\}$