[Reviews · NeurIPS 2019]

Reviewer 1



This work presents ways to obtain estimates of the aleatoric and epistemic uncertainties of deep neural networks. The aleatoric uncertainty is estimated by learning the quantiles of the target variable via Simultaneous Quantile Regression (SQR); it minimizes the pinball loss where the target quantile is randomly sampled in every training iteration. The epistemic uncertainty is implicitly estimated by Orthonormal Certificates (OCs); these are functions that are trained to map in-distribution examples to zero whereas out-of-distribution examples to non-zero values. The authors also provide tail bounds for the OCs in the case of Gaussian input data, which does provide some intuition about the behaviour. Simplicity is a benefit of these estimators and the authors demonstrate their performance on regression and classification tasks. Uncertainty estimation for neural networks is an important topic and relevant for Neurips. The paper itself is well written, and conveys the main ideas in an effective manner. The ideas themselves are interesting and, as far as I am aware, novel. They seem to be simple to implement, a fact that gives them a nice bonus for practical applications. The experimental evaluation is relatively thorough as there is a comparison to a variety of other methods. My main point of feedback is that the authors tend to exaggerate their results, as they claim SOTA performance. - For the evaluation of the aleatoric uncertainty they experiment on the regression task from [1] and their results show that SQR is usually within the standard deviation of the much simpler Conditional Gaussian baseline, thus offering no real improvement. Granted that the predictive distributions might not be very complicated to begin with in those datasets (as a single Gaussian works just as well), the authors should find a different task to make the effectiveness of SQR more convincing. Furthermore, as SQR promises more flexible predictive distributions (e.g. asymmetric, multimodal etc.), I believe that the authors should also compare with a mixture of Gaussians likelihood, another simple alternative that can facilitate for such predictive distributions. - For the evaluation of the epistemic uncertainty the authors compare against various methods on an out-of-distribution detection task. Similarly to the previous point, the performance of OCs is usually within the standard deviation of the baselines thus denoting that OCs do not provide a (consistent) significant improvement. Furthermore, for the svhn dataset it seems that the bold number should instead go to “largest” and “entropy” rather than OCs? They both have a 0.93+-0.01 compared to the 0.91+-0.00 of OCs. As for other comments for general improvement of the manuscript: - For the regularization of OCs the authors seem to employ a soft constraint for orthonormality; while this is simple to implement it introduces one extra hyperparameter that needs tuning, the regularizations strength \lambda. How sensitive are OCs to the setting of \lambda? A way to bypass this hyperparameter is via a hard constraint. Did you experiment with such a constraint, e.g. [2] and related, for the certificates? - What was the network architecture that you used for the aleatoric uncertainty experiment and what was RMSE for all of the methods? Furthermore, what was the dropout rate that you employed and did you optimize over it (e.g. with cross-validation)? Regression tasks are usually more sensitive to the dropout rate than classification ones. - What was the choice of the loss for the OCs in the epistemic uncertainty experiment? It is a bit unclear to get this information from the code as there are two possible choices. - Why did you select a part of the training set as an out-of-distribution dataset? While I do agree that it is more challenging, it also makes the comparison against other works more difficult, as this is not the standard evaluation. Furthermore, I believe that it will be worthwhile to have at least one experiment where you compare out-of-distribution detection across-datasets. For example, you can take the model trained on “CIFAR-5” and then evaluate the detection from the baselines and OC’s on SVHN (similarly for MNIST and Fashion-MNIST). This will demonstrate whether each method generalizes across the peculiarities between datasets. Overall, I believe that this is good work and, if the authors address my comments, I will recommend for acceptance. [1] Probabilistic Backpropagation for Scalable Learning of Bayesian Neural Networks [2] Efficient Orthogonal Parametrisation of Recurrent Neural Networks Using Householder Reflections # Update after the authors rebuttal I am happy to increase my score, since the authors adequately addressed my comments in their response.

Reviewer 2



The paper describes methods for estimating aleatoric and epistemic uncertainties in a neural network. While many existing methods are based on sampling based approaches, the paper proposes a sampling-free method. The aleatoric uncertainty is estimated using quantile regression based on the pinball loss. The pinball loss had been used for quantile estimation. But I am not certain if it has not been used in a neural network before. The epistemic uncertainty is estimated by the proposed orthonormal certificates, which finds sort of the null space of training data points. - Overall, there are some interesting aspects about the paper but it is not convincing enough to support its acceptance. - The benefits of the proposed method are not fully illustrated in experiments. - Synthetic datasets can be used to show convergence of the proposed method and its sample efficiency against competing methods. - In experiments, reporting the training times and inference times can help position the proposed method against others. - Recently, there are some sampling-free methods for estimating epistemic uncertainty. But they are not compared in the paper. - On aleatoric uncertainty estimation, the paper proposes to use the pinball loss. Since the loss function is different from the popular loss function, such as cross entropy, it is important to verify if the use of the proposed loss does not hurt the performance on a large model, such as ImageNet (not UCI datasets). - On epistemic uncertainty estimation, the paper assumes that the feature space is linear. This fact needs to be more elaborated as not all output feature spaces of neural networks are vector spaces. For the loss function l_c in (4), the investigation of the effects of different norms, other than l2 used in the paper, will be interesting. Minor: - The dataset used in 4.1 is not mentioned. - An incorrect legend on the right most figure of Fig. 2.

Reviewer 3



Summary: The authors consider the task of predicting Y (the continuous variable is the most studied in this work ) given a set of features X by fitting a model. The aim is to provide uncertainties about the prediction. In the case of Regression, confidence intervals need to be estimated. Therefore, authors propose to use a pinball loss function with a quantile threshold tau, that in expectation is the tau-th quantile of the conditional distribution Y given X. Authors propose a variant called Simultaneous quantile regression when they uniform randomly sample the the quantile threshold tau to optimize for. This results in a simultaneous optimization of all quantiles through the randomized pinball loss. I think the randomized version is novel although pinball loss has been used before. The authors demonstrate how a model trained using pinball loss helps in estimating the 95% prediction interval better than many competitors. Conditional Gaussian fitting with its variance providing estimate (although one of the simpler baseline) is the next reasonable close competitor. I find the numerical results quite exhaustive with many methods compared. The authors then move onto the filtering problem (they call it epistemic uncertainty) when authors propose finding k orthonormal certificates c_i (each is a vector) that lies close to the null space or a subspace with very low eigenvalues of the covariance matrix of some high level feature vector in an already pretrained neural network. So essentially the score is the average of the sum of dot product between these certificates at the feature vector and evaluates to close to 0 for points lying in the training distirbution. They show that if the features are Gaussian with some covariance, how the score is a pretty good separator of out of sample and in-sample points where out of sample has a mean shift and a different covariance. They demonstrate by comparing it various methods for the filtering task how they get state of the art or competitive with the state of the art on CIFAR-10, MNIST, Fashion-MNIST etc.. Out of samples are those data points in one half of the classes and in-sample are those samples belonging to the other half of the classes . Authors further show numerical evidence of the SQR actually doing pretty well on detecting causal directions between a pair of variables. Although unrelated, it seems like the pinball loss is a good measure of complexity of a conditional distribution. Therefore, the direction with the lowest randomized pinball loss is chosen as the correct direction. Even here the empirical demonstrations are comprehensive. Originality : Although the pinball loss is strongly motivated by prior work, randomized version of that simultaneously optimizing for all quantiles is novel and seems to have very competitive empirical performance on a variety of tasks. The OC approach is very novel for the filtering task. Quality: Technically it is sound. However, given the formulation of Orthonormal certificate "approximately" tracing the null space of the feature covariances - whats the need for Theorem 1 ? It just solves the Gaussian case anyways and is not surprising. I think including the application of SQR for causality in the main paper would enhance the paper. Theorem 1 adds little and could be moved to the appendix as an intuitive formal justification for OCs. Clarity: The motivations behind OCs and SQRs are clear. However, the application of SQR methods to classification is not clear. So I have some questions here: a) Lines 92-94 "When dealing with multivariate classification it is possible to proceed similarly (build one SQR on 93 top of each logit), or use calibration methods [29] to interpret softmax scores as aleatoric uncertainty 94 estimates" - If existing calibration methods like temperature scaling are to be used as in ref 29, where is the novelty for classification. Except the case of ordinal targets discussed in Appendix B, it is not clear what this one line "build SQR on top of each logit" means ? Why are there no systematic experiments for classification setup ? I find this is a wekaness of the paper. The authors might as well drop this and say that their method is meant for regression mostly. b) Lines 165 - 67 says that when |x| goes to infinity upper bound goes to infinity, why is that an evidence that |C^T x| will also go to infinity. At least the point is not clear and the exact technical statement of the quantity going to infinity if the upper bound goes to infinity seems a bit awkward reasoning. c) There is a lot of recent work on the filtering task of rejecting samples in the classification setting - https://arxiv.org/pdf/1705.08500.pdf, https://arxiv.org/pdf/1805.05396.pdf. Further there is some more work on uncertainty estimation (related to the SQR algorithm) based on observing confidences of the model during various snapshots of the training process - https://arxiv.org/pdf/1805.05396.pdf. These works must be cited and their differences discussed. Significance: Both problems tackled are real and very important. The new methods could act as strong baselines for future work. The methodological novelty is also very interesting. Weaknesses: I have listed weaknesses primarily in Clarity and Quality sections. Mainly, I am not sure of the authors assurances of being able to extend SQR to the classification setting. It seems only ideally suitable for regression and possibly when target is finite and ordinal. Some key citations are missing. Since the authors did comprehensive evaluation on a large set of baselines and proposed novel methods, I would not press on new comparisons. However, they should discuss these recent relevant pieces of literature. ***After rebuttal ****** The authors have reported a number of new experiments and their results. They have also adequately addressed my concerns. So I am fine with the current score.

[Author Response · NeurIPS 2019]

We thank the reviewers for their valuable feedback. We address your concerns as outlined below.

**Reviewer #1**

• *Claims about SOTA performance.* We toned down the discussion of our experimental results. We now claim
"competitive performance", and rely on the simplicity of SQR/OC to establish them as "new and strong baselines".

• *Performance of SQR.* The confidence intervals of SQR are slightly better calibrated than ConditionalGaussian, while
being narrower. However, we agree that SQR shines when dealing with multi-modal uncertainties. We now illustrate
this on real data from the Old Faithful geyser in Yellowstone National Park, Wyoming, USA.

• *Performance of OCs.* We now report the average accuracy of out-of-distribution detection across the four presented
datasets for all methods. OCs are the best performing with an average accuracy of $90\%$. The second best performance
is obtained by the entropy estimator, at $87\%$.

• *Performance of OCs across datasets.* We believe that OOD must be powerful enough as to reject images with the
same statistics as the training data, but different classes. We now also report an experiment where we measure OOD
performance across datasets. OCs remain the best performing method. For example, OCs are able to use an MNIST
network to reject 96% of Fashion-MNIST examples, followed by ODIN at 85%.

• *Sensitivity wrt $\lambda$.* We now report an ablation study showing that OCs are stable with respect to the choice of $\lambda$. Our
ablation shows that any value greater than 10 achieves competitive results. Looking at a hard-constrained version of
OCs is an exciting direction for future work.

• *Neural network architectures and dropout cross-validation.* We now specify all the neural network architectures in
the Appendix (for SQR, two layers of 64 ReLU units). We now cross-validate the value of dropout for all experiments
in $\{0.1, 0.25, 0.5, 0.75\}$, obtaining very similar results to the ones reported in our submission.

• *Loss choice for OCs.* We now emphasize the text (L133-135) explaining that we use the same loss as the one
employed to train the entire network. In this case, the cross-entropy loss.

• *Report MSE in SQR experiments.* The only method trained to optimize MSE is ConditionalGaussian. SQR (as well
as the other quantile-based methods) optimizes MAE (at the median). If minimizing MSE is important for the task,
we propose having an extra output in the neural network dedicated to that issue.

**Reviewer #2**

• *Benefits and performance of the proposed methods.* Please see the first four items in the response to Reviewer #1.

• *Sample efficiency.* We now leverage results from the literature on quantile regression and principal component
analysis to discuss the (optimal) sample efficiency of SQR and OC.

• *Comparison to sampling-free methods.* We now discuss (Hwang et al., 2018; Le et al., 2018; Feng et al., 2019).

• *Test SQR on ImageNET.* We now refrain from talking about classification in the SQR section, and focus the application
of SQR to regression problems. Therefore, SQR is not applicable to ImageNET. However, please note that SQR can
be implemented as an extra output neuron, therefore not impacting the original accuracy of the network.

• *Linearity assumption on OCs.* We now emphasize the text (L130-132) describing how to construct non-linear OCs.
In any case, restricting our experiments to linear OCs is justified by the manifold hypothesis (Bengio et al., 2013),
which states that deep learning models attempt to arrange the data manifold to linearly separate the examples.

• *Choice of $l_c$.* We now emphasize the text (L134-135) discussing the choice of $\ell$, and added a small experiment in the
Appendix that compares the cross-entropy (superior) to the MSE loss.

**Reviewer #4**

• *Theorem 1 and causality experiment.* We have now moved Theorem 1 to the Appendix, and the causality experiment
to the main text.

• *SQR and classification.* We now refrain from applying SQR to classification, and focus on regression problems.

• *Behaviour of $\|C^\top x\|$.* We thank the reviewer for pointing this out this subtle mistake. We have removed the
explanatory paragraph (L165-167). Now we rely on Theorem 1 to describe (in a more precise manner) the behaviour
of $\|C^\top x'\|$ as $x'$ moves far away from the nullspace of the training data.

• *Missing references.* We now discuss (Geifman and El-Yaniv; Chen et al) in our related work section.

[Meta-Review · NeurIPS 2019]

The paper presents an approach for estimating aleatoric uncertainty which leverages the pinball loss in quantile regression, and orthonormal certificates for measuring epistemic uncertainty. Although the pinball loss has been used in prior work, its randomized version for simultaneously optimizing for all quantiles is novel. In addition the novelty of the OC approach for the filtering task is significant. Overall the value of the proposed methods is convincingly demonstrated on a variety of datasets. The reviewers and AC have carefully examined the author feedback and feel that the feedback adequately addresses the concerns raised in the reviewers. We strongly encourage the authors to incorporate their feedback and in particular their new experiments as these would strengthen their submission significantly.